# Sitting Posture, Sagittal Spinal Curvatures and Back Pain in 8 to 12-Year-Old Children from the Region of Murcia (Spain): ISQUIOS Programme

**DOI:** 10.3390/ijerph17072578

**Published:** 2020-04-09

**Authors:** Pilar Sainz de Baranda, Antonio Cejudo, María Teresa Martínez-Romero, Alba Aparicio-Sarmiento, Olga Rodríguez-Ferrán, Mónica Collazo-Diéguez, José Hurtado-Avilés, Pilar Andújar, Fernando Santonja-Medina

**Affiliations:** 1Department of Physical Activity and Sport, Faculty of Sport Sciences, Regional Campus of International Excellence “Campus Mare Nostrum”, University of Murcia, 30100 Murcia, Spain; psainzdebaranda@um.es (P.S.d.B.); mariateresa.martinez13@um.es (M.T.M.-R.); alba.aparicio@um.es (A.A.-S.); olga.rodriguez@um.es (O.R.-F.); 2Sports and Musculoskeletal System Research Group (RAQUIS), University of Murcia, 30100 Murcia, Spain; monicacodi@hotmail.com (M.C.-D.); joseaviles@um.es (J.H.-A.); pilarandujar.albacete@gmail.com (P.A.); fernando@santonjatrauma.es (F.S.-M.); 3Department of Rehabilitation Sciences and Physiotherapy, Albacete University Hospital Complex, 02006 Albacete, Spain; 4Department of Medicine and Orthopaedic Surgery, Faculty of Medicine, University of Murcia, 30100 Murcia, Spain

**Keywords:** sagittal morphotype, assessment, spinal imbalance, spinal curvatures, spine, asthenic sitting, postural hygiene, postural education, scholar, slump sitting

## Abstract

To explore sagittal spinal alignment and pelvic disposition of schoolchildren in a slump sitting position is needed in order to establish preventive educational postural programs. The purposes of this study were to describe sagittal spinal alignment and pelvic tilt (LSA) in a slump sitting position and to explore the association of sagittal spine and pelvic tilt with back pain (BP) among 8–12-year-old children. It was a cross-sectional study. Sagittal spinal curvatures, BP and pelvic tilt were assessed in 582 students from 14 elementary schools. It was found that 53.44% of children had slight thoracic hyperkyphosis and that 48.80% presented moderate lumbar hyperkyphosis and 38.66% presented slight lumbar hyperkyphosis. Those who did not suffer from BP in any part of the back had a higher lumbar kyphosis (24.64 ± 7.84) or a greater LSA (107.27 ± 5.38) than children who had some type of BP in the previous year or week (lumbar kyphosis: 23.08 ± 8.06; LSA: 105.52 ± 6.00), although with no clinically relevant differences. In fact, neither sufferers nor those who did not have BP presented normal mean values for lumbar kyphosis or LSA according to normality references. This study demonstrates the need to assess sagittal morphotype in childhood since schoolchildren remain incorrectly seated for many hours and it greatly affects their spinal curvatures.

## 1. Introduction

The study of the student’s posture while sitting in schools is of great importance due to its influence on spinal development and maintenance [1,2]. Furthermore, spinal morphology is greatly affected by acquired and sustained postures in seated positions [2,3,4].

Sagittal spinal alignment in the sitting position is different than that in the standing position and it changes as the child grows [5]. When sitting, there is a posterior tilt of the pelvis and a verticalization of the sacrum [6,7], which results in a more or less pronounced loss of the lumbar lordosis with regard to standing [6,8,9,10,11,12]. Under these conditions, the intervertebral discs can suffer compression in the anterior part of the fibrous ring, a posterior migration of the nucleus [13,14], and an increase in the intradiscal pressure [13,15], which causes damage to the intervertebral discs [16] and increases the tension in the passive elements of the posterior part of the spine [17]. 

Nowadays, children usually spend many hours in a seated posture. In fact, they usually spend around 60% and 80% of their school day in a sitting position [18,19]. Additionally, the time that is spent on homework and activities such as watching TV, using the computer, and playing on a tablet can be added to that seated time. Therefore, one of the biggest postural problems is slump sitting position, which is very common among schoolchildren [20].

In comparison with “neutral” sitting position, slump sitting position increases the compressive forces on the spine [8,21]. This concentration of stress can produce degeneration [22]. Thus, a correct sitting posture is fundamental for the adequate development of the sagittal spine. 

Within the school environment, one of the reasons for slump sitting is the use of furniture that is not adapted to the anthropometry of schoolchildren [23,24]. It is common for students of different ages and notably different heights to utilize the same-sized desks with fixed dimensions. It should be noted that Quintana-Aparicio et al. [25] already demonstrated how students between 8 and 10 years of age sat in hyperkyphosis in more than half of the cases when listening to the teacher. It is important because if the slump is continued for a long time, it may cause pain and diseases at the cervicothoracic and lumbosacral joints [26]. Slump sitting position is associated with an increase in the spinal load [8,21], and there is greater pressure on the intervertebral disc than in the neutral posture [27,28] and a concentration of stress that can produce degeneration [22,29,30]. 

For that reason, to explore sagittal spinal alignment and pelvic disposition of schoolchildren in a slump sitting position is needed in order to establish preventive educational programs which can address postural misalignments from an early age to prevent back pain (BP).

On the other hand, certain studies have observed that sedentary behaviors are an important factor in the development of BP during adolescence [31]. Indeed, the sitting position is recognized as being the main aggravating factor of low back pain (LBP) [27,32]. Nevertheless, previous literature has concluded that there is no clear evidence to support that there is a relationship between poor school posture and the development of BP [33,34]. Thus, the actual influence of sagittal spinal misalignment while sitting on BP remains unclear and needs to be further investigated.

Therefore, the purposes of this study were (a) to describe sagittal spinal alignment and pelvic tilt in a slump sitting position as well as (b) to explore the association of sagittal spine and pelvic tilt with BP among 8–12-year-old children.

## 2. Materials and Methods 

### 2.1. Design

It was a cross-sectional study. Prior to participation in an exercise program on posture and physical fitness during physical education classes (ISQUIOS Programme), angular values for sagittal spinal curvatures (thoracic and lumbar) of elementary school students were recorded in a slump sitting position. Sagittal spinal morphotype was analyzed by sex, age and regarding the BP suffered in the previous week and in the preceding year.

### 2.2. Participants

Firstly, a total of 887 students were selected through a convenience sample from 16 elementary schools that had been selected to participate in the ISQUIOS Programme, a postural educational program which is carried out in the Region of Murcia (Spain), and volunteered to participate in the study. All measurements were performed prior to the postural education program.

As inclusion criteria, those who were in 3rd–6th grade and were from 8 to 12 years old (a), who attended the day of the assessment (b) and who delivered the signed written consent (c) were included (n = 812). However, those who had suffered an important physical injury which limited the correct performance of the test (a), who had been previously diagnosed to have scoliosis or who had previously received treatment for any frontal or sagittal plane-related pathology (b) or those who did not complete the BP questionnaire (c), were excluded (n = 220). In addition, some cases were excluded due to missing data for one of the variables (pelvic tilt, thoracic angle or lumbar curve angle) (n = 10). Furthermore, the statistical analysis was performed with 582 students from 14 elementary schools (Figure 1). 

The study was conducted in accordance with the Declaration of Helsinki, and the protocol was approved by the Ethics and Research Committee of the University of Murcia (Spain; Protocol Number 77/2013). Therefore, all the students and legal tutors were informed of the procedure and objectives of the study and signed a written consent form.

### 2.3. Procedures

Students were instructed not to participate in any training or physical activity 24 h before their assessment. All the measurements were performed on the same day, starting with anthropometric measurements. Body height was measured with Seca 213 mobile stadiometer, with an accuracy of 0.1 cm. Body mass was measured using electronic scale OMRON BF 500, with an accuracy of 0.1 kg. Data from each student were obtained during the same session. Measurements were performed by the same physician expert and participants were assessed wearing undergarments and barefoot. Students did not perform warm-up or stretching exercises before or during the measurement [35,36]. Three trials for each measure were administered/recommended. When two of those measures were equal, we chose that value. When the three measures were different, we took the average value of the two similar measurements for data analysis. Furthermore, it is very important to note that a test trial was carried out before the first measurement with the objective that the student was well informed and was absolutely sure about how to perform the test. The measurements were made in a randomized order (thoracic and lumbar sagittal curves and pelvic tilt). The study was rigorously controlled by keeping the expert and the students blinded to the objective of the study.

#### 2.3.1. Thoracic and Lumbar Sagittal Curves Assessment

An unilevel inclinometer (ISOMED, Inc., Portland, OR) was used to quantify the sagittal spinal curvatures by providing considerable reproducibility and validity, with a good correlation with the radiographic measurement [37]. In order to establish the reliability of the examiner, a double-blind study with 10 subjects was performed before the assessment, with an intraclass correlation coefficient (ICC) of 0.95 for the thoracic kyphosis and 0.95 for the lumbar lordosis.

The protocol described by Santonja [38] was used to assess the sagittal spinal disposition, which has been previously used in other studies [35,36,39,40]. Prior to data collection, the spinous process of the first thoracic vertebra (T1), the lumbar-thoracic transition (T12-L1) and the fifth lumbar vertebrae (L5) were marked on the skin of participants [41]. In order to assess the slump sitting position, the participant sat on a stretcher in a relaxed position and without feet support (with the eyes and ears in line horizontally, forearms placed over the thighs and knees flexed) [35,36].

First, the inclinometer was placed on the first mark (T1), and it was calibrated to 0°. Then, the inclinometer was placed on the second mark (T12-L1), and the grades for the thoracic curve were recorded. Subsequently, at this point, the inclinometer was calibrated to 0° again to be situated on the third mark (L5-S1) in order to assess the lumbar curve (Figure 2). The legs on the inclinometer were adjusted to cradle the spinous processes and were pressed gently but firmly into the interspinal spaces.

The values described by Santonja [42] were used to classify the results related to the angular values of the thoracic and lumbar curvature in a slump sitting position. The references of normality for thoracic curve were: reduced kyphosis or hypokyphosis <20°; normal kyphosis: 20° to 40°; slight hyperkyphosis: 41° to 60°; moderate hyperkyphosis >60°. As for lumbar curvature, the following categories were used: hypokyphosis <−15°; normal kyphosis (Normal lumbar curvature): −15° to 15°; slight hyperkyphosis: 16° to 25°; moderate hyperkyphosis >25°. Negative values corresponded to lumbar lordosis (posterior concavity) and positive values corresponded to lumbar kyphosis.

#### 2.3.2. Pelvic Tilt Evaluation in a Slump Sitting Position

In order to assess pelvic tilt, the lumbosacral angle (LSA) in a slump sitting position was measured [43,44]. The study of LSA is of great interest as it indicates if participants were able to keep their pelvic verticality, and thus, a more neutral sagittal spine in a slump sitting position [43,44].

A goniometer was used to perform this measurement. The branches of the goniometer were aligned with the horizontal line and the spinous processes of L4-S1 in order to record the angle between the two references; however, the supplementary angle was used for the data analysis (Figure 3). The following values were used as references of normality [45]: normal: ≤100°; slight retroversion (posterior pelvic tilt): 101° to 110°; significant retroversion or posterior pelvic tilt: >110°.

#### 2.3.3. Back Pain Assessment

An ad-hoc questionnaire composed of 8 questions and based on previous studies was employed to describe BP prevalence in schoolchildren [46,47,48,49].

The survey starts by asking questions based on socio-demographic issues such as sex, age, school, course and pathologies. After that, it is asked about BP suffered during the preceding year or week. BP was defined as aching, pain or discomfort in some part of the back that was not related to trauma or menstrual pain [46]. The questionnaire included a drawing of the back to mark the back area where the pain was suffered (Figure 4). If the back pain in the previous year was suffered as a one-time event or was experienced more than once was also evaluated.

The questionnaire was completed by parents. An experienced researcher presented the questionnaire to the parents and explained the procedure to complete the survey in accordance with the students’ responses to all the questions.

### 2.4. Statistical Analyses

Data analysis was conducted using SPSS v.24 (IBM, Armonk, NY, USA). Descriptive statistics including mean values and standard deviations (SD) for the total population, as well as by gender and age groups, were performed. 

Pairwise comparison of means (Student t-test for independent samples) was used to examine spinal and pelvic tilt differences between genders and between those who had or did not have BP. In addition, the percentages of children with a normal or an imbalance spine were calculated, and the Pearson chi-squared test was used to examine the differences in the percentages of normality and sagittal imbalance between gender and age groups as well as between pelvic tilt categories. Furthermore, one-way analysis of variance (ANOVA) was carried out to analyze height and weight among spinal curvatures and pelvic tilt categories and to examine spinal curves and pelvic tilt by age group. Post-hoc comparisons were performed through Bonferroni. Pearson intraclass correlation coefficient (ICC) and “r of Pearson” were calculated to analyze the association between sagittal spinal curves and pelvic tilt.

A backward stepwise binary logistic regression was used to identify the relationship of sagittal spinal curvatures and pelvic tilt with BP (forward selection [conditional], inclusion probability *p* ≤ 0.05, elimination probability *p* ≤ 0.10). The resulting odds ratios (ORs) and associated 95% confidence intervals (CIs) were reported.

To determine whether it is possible to find a clinically relevant cut-off point for the lumbar curvature that can be used for pointing out individuals at a high risk of developing LBP, receiver operating characteristic (ROC) curves were calculated. The area under the ROC curve represents the probability that a selection based on the risk factor for a randomly chosen positive case will exceed the result for a randomly chosen negative case. The area under the curve can range from 0.5 (no accuracy) to 1.0 (perfect accuracy). If it is found to be statistically significant, it means that using the risk factor as a determinant is better than guessing. Since the ROC curve plots sensitivity against 1 minus specificity, the coordinates of the curve can be considered possible cut-off points, and the most suitable cut-off can be chosen. 

Eta-square (η^2^), “d” of Cohen and OR statistics were calculated to determine the effect size [effect sizes for the OR were defined as follows: small effect OR = 1–1.25, medium effect OR = 1.25–2 and large effect OR ≥ 2 [50]. The level of significance was set at *p* < 0.05.

## 3. Results

Table 1 shows the demographic data of the participants who were included and analyzed in the present study.

### 3.1. Sagittal Spinal Alignment and Pelvic Tilt in a Slump Sitting Position

As Table 2 shows, mean angular values for thoracic and lumbar curves were 42.70 ± 9.34° and 24.29 ± 7.91°, respectively. The mean pelvic angle while sitting was 106.88 ± 5.57°. Significant differences were found when spinal curvatures and pelvic tilt were compared by sex. Males showed a higher thoracic [t(581) = 2.915, *p* = 0.004, d = 0.24] and lumbar kyphosis [t(581) = 4.836, *p* < 0.001, d = 0.40] than females, as well as a higher LSA [t(581) = 4.752, *p* < 0.001, d = 0.39] than females.

In addition, significant differences were found by age with respect to thoracic angle [F(4,578) = 4.561, *p* = 0.001, η^2^ = 0.031], lumbar angle [F(4, 578) = 3.195, *p* = 0.013, η^2^ = 0.0216] and pelvic tilt [F(4,578) = 7.281, *p* < 0.001, η^2^ = 0.0479]. Concretely, 10-year-old children had a higher thoracic angle (*p* = 0.007; d = 0.36) and lumbar angle (*p* = 0.047; d = 0.29) than 11-year-old children, while 12-year-old children presented a greater thoracic curve than 11-year-old children (*p* = 0.013; d = 0.57). Furthermore, 11-year-old children had a smaller LSA than 8-year-old (*p* < 0.001; d = 0.62), 9-year-old (*p* = 0.001; d = 0.46) and 10-year-old children (*p* < 0.001; d = 0.46).

Table 3 shows that most of the students had slight thoracic hyperkyphosis (53.44%) or normal thoracic kyphosis (44.50%). Likewise, it is shown that 48.80% of children had moderate lumbar hyperkyphosis and that 38.66% presented slight lumbar hyperkyphosis. On the other hand, 68.21% of children were diagnosed with a slight pelvic retroversion or a mild posterior pelvic tilt.

When the results were compared by gender, male sex was significantly associated to slight thoracic hyperkyphosis, while female sex was significantly associated to have a normal thoracic kyphosis [X^2^(582) = 8.315, *p* = 0.040, η^2^ = 0.115]. As for the lumbar curve, male sex was significantly associated to have moderate lumbar hyperkyphosis, while female sex had a significant association with normal lumbar kyphosis and slight lumbar hyperkyphosis [X^2^(582) = 25.736, *p* < 0.001, η^2^ = 0.209].

When pelvic tilt was analyzed by gender, male sex was negatively associated with having a normal pelvic tilt, whereas female sex was associated to a normal pelvic retroversion while sitting [X^2^(582) = 11.521, *p* = 0.003, η^2^ = 0.133].

Thoracic morphotype had no significant relationship with age [X^2^(582) = 19.535, *p* = 0.076]. When lumbar morphotype was analyzed by age, it was found that 11-year-old children had a significant association with normal lumbar kyphosis, whereas 8-year-old children presented a negative association with having a normal lumbar kyphosis [X^2^(582) = 25.296, *p* = 0.001, η^2^ = 0.158]. In this sense, when the status of pelvic tilt was compared by age, it was found that the 11-year-old group had a positive association to normal pelvic tilt and a negative relationship with a significant pelvic retroversion while sitting [X^2^(582) = 23.116, *p* = 0.003, η^2^ = 0.165].

When height and weight were compared between thoracic categories, no differences were found [(F(4,578) = 1.429, *p* = 0.233) and (F(4,578) = 1.509, *p* = 0.211), respectively]. However, significant differences were found when height was analyzed by lumbar categories or pelvic tilt [(F(4, 578) = 10.852, *p* < 0.001, η^2^ = 0.070) and (F(4, 578) = 13.010, *p* < 0.001, η^2^ = 0.083), respectively]. It was determined that those with a normal lumbar kyphosis were significantly taller than those with slight lumbar hyperkyphosis (*p* = 0.003; d = 0.46) or moderate lumbar hyperkyphosis (*p* < 0.001; d = 0.60). Likewise, students with a normal pelvic tilt were also significantly taller than those who had a slight pelvic retroversion (*p* < 0.001; d = 0.53) or a significant retroversion (*p* < 0.001; d = 0.66).

In addition, significant differences were found when weight was analyzed by lumbar categories (F(4,578) = 56.723, *p* < 0.001, η^2^ = 0.282) or pelvic tilt (F(4,578) = 74.483, *p* < 0.001, η^2^ = 0.340). Children with a normal lumbar kyphosis were significantly weightier than those who had slight lumbar hyperkyphosis (*p* < 0.001; d = 0.81) or moderate lumbar hyperkyphosis (*p* < 0.001; d = 1.36). In this sense, students with slight hyperkyphosis were significantly weightier than those with moderate hyperkyphosis (*p* < 0.001; d = 0.51). Furthermore, children with a normal pelvic tilt were also significantly weightier than those with a slight retroversion (*p* < 0.001; d = 1.30) or a significant retroversion (*p* < 0.001; d = 1.44), as well as students who had slight pelvic retroversion, were weightier than those who presented a significant pelvic retroversion (*p* < 0.013; d = 0.35).

On the other hand, a positive significant relationship was found between thoracic angle and pelvic tilt, which means that the greater the thoracic curve is, the higher the pelvic retroversion is found (r = 0.291; *p* < 0.001). A positive significant correlation was also determined between lumbar kyphosis and pelvic retroversion (r = 0.638; *p* < 0.001).

As it is represented in Table 4, there was a statistically significant association of thoracic (X^2^(582) = 34.866, *p* < 0.001, η^2^ = 0.231) and lumbar kyphosis (X^2^(582) = 165.064, *p* < 0.001, η^2^ = 0.474) with pelvic tilt categories.

### 3.2. Sagittal Spinal Alignment in a Slump Sitting Position, Pelvic Tilt While Sitting and Back Pain

In Table 5, it is shown that those schoolchildren who suffered from BP in the previous year had a smaller kyphotic lumbar curve (23.08 ± 8.06) than those who did not have BP in the preceding year (24.64 ± 7.84) [t(580) = 1.990, *p* = 0.047, d = 0.20]. On the other hand, students who did not present BP, mid-BP or LBP in the previous year, had a greater LSA than sufferers [(t(580) = 3.186, *p* = 0.002, d = 0.32), (t(80) = 2.497, *p* = 0.015, d = 0.37) and (t(574) = 2.017, *p* = 0.044, d = 0.30), respectively]. In addition, those who had upper BP in the previous week presented less lumbar kyphosis than non-sufferers [t(576) = 2.127, *p* = 0.034, d = 0.68]. As for the recurrence, those who experienced recurrent back pain had similar angles for spinal curves and pelvic tilt than those who suffered BP as a one-time event in the preceding year (*p* > 0.05).

However, no association was observed when BP in the previous year and upper BP in the preceding week were analyzed by categories of lumbar curvature according to normality references (*p* > 0.05). Likewise, no significant relationship was detected when BP, middle BP and LBP in the previous year were analyzed by categories of LSA according to normality references (*p* > 0.05).

With the stepwise logistic regression analysis, it was found that the LSA was only slightly associated with BP in the previous year (OR = 1.059 [small]; 95%CI = 1.021 to 1.097, *p* = 0.002), with mid-back or thoracic pain in the previous year (OR = 1.069 [small]; 95%CI = 1.021 to 1.118, *p* = 0.004), as well as with LBP in the previous year in the assessed children (OR = 1.054 [small]; 95%CI = 1.001 to 1.109, *p* = 0.045). Likewise, the angle for lumbar curvature had only a small effect on the cervical pain suffered in the previous week (OR = 1.082 [small]; 95%CI = 1.005 to 1.116, *p* = 0.036) (Table 6).

The areas under the ROC curve demonstrated that either LSA or lumbar angle had a poor predictive model accuracy [51] for any type of BP (Figure 5).

Finally, when the chi-square test was performed, it was confirmed that having a greater or a lower lumbar curvature or LSA (according to the cut-off points found) was not associated with any type of BP (*p* > 0.05).

## 4. Discussion

The first purpose of the present study was to describe sagittal spinal alignment and pelvic tilt in a slump sitting position among 8 to 12-year-old children. It was found that 53.44% of children had a slight thoracic hyperkyphosis and that 48.80% presented moderate lumbar hyperkyphosis, as well as 38.66% were diagnosed with slight lumbar hyperkyphosis in a slump sitting position. However, only 44.50% of students presented a normal thoracic kyphosis, as well as only 12.54% had a normal lumbar kyphosis while sitting. 

In this sense, it was also observed that only 14.78% of students presented a normal pelvic disposition, while 68.21% of children had a slight pelvic retroversion when the pelvic tilt was valued through the LSA in a slump sitting position. These results confirm that most of the students present an incorrect and kyphotic sitting posture with the pelvis tilted posteriorly.

It must be noted that an incorrect posture, particularly while sitting and prolonged sitting, has been considered to be the most important factor for the appearance and maintenance of LBP, as well as strain in the neck [52,53]. The loss of lordosis and protracted head are the most common consequences of stress while sitting [27,54,55,56].

The combined effect of lumbar and dorsal hyperkyphosis, as well as the protracted head, decrease the mechanical resistance for anterior shear forces [57,58,59]. This can give rise to the formation of wedges due to the compression of the growth nuclei during puberty [38], as well as inversions of the intradiscal space and increased pressure on the discs.

One of the problems of this posture is that in the majority of cases, it is relatively comfortable, since the muscles do not need to contract, and therefore produces a sense of relaxation [20]. However, an incorrect sitting posture can eventually give rise to lower-back problems, due to several reasons: (a) increased anterior shear stress; (b) increased pressure in the anterior side of the vertebrae, making them more prone to wedges; (c) excessive pressure on the intradiscal space, which increases the pressure on the anterior part of the disc and fibrous ring. This causes a posterior movement of the nucleus pulposus that pushes onto the wall of the fibrous ring, with a risk of protrusion and possibly damaging the ring. Finally, (d) increased tension in posterior ligaments, especially due to muscle relaxation [20,21,26]. 

As kyphosis increases, the joints receive less weight, and more stress is applied to the ring and nucleus pulposus. This, in turn, results in the spine being less resistant to compressive loads and could cause the sinking of the vertebral plate into the trabecular bone. Sustained pressure on the spine in hyperkyphosis reduces its resistance to the loads and decreases tension in the ligaments, possibly increasing disc protrusion [58]. Furthermore, the vertebral segments submitted to a compressive load while maintaining a hyperkyphotic posture are less resistant (43%–47% less) to fractures [58].

Therefore, an incorrect sitting posture increases the stress and pressure on the discs and contributes to its degeneration and sense of pain. The risk factors of this posture are static load on the muscles and inversion of the lumbar curvature [58].

For that reason, Gunning et al. [58] highlight the importance of measuring sagittal spinal alignment and pelvic disposition and the relevance for the schoolchildren population, as they remain seated for many hours at school and home. Therefore, they foreground that the early detection of incorrect sitting postures is essential in order to apply treatment or prevention methods. It is to this end that the analysis of the spinal morphotype while seated should be part of the spinal assessments performed on a regular basis in students.

In the current study, when spinal curvatures and LSA were analyzed by sex, it was determined that males presented higher thoracic and lumbar kyphotic curves as well as pelvic retroversion while sitting than females. In fact, males were more prone to present slight thoracic hyperkyphosis or moderate lumbar hyperkyphosis than females, whereas significantly more females than males had a normal pelvic tilt.

In accordance with our results, O’Sullivan, Smith, Beales, and Straker (2011) [60] found in their study of sitting posture with adolescents that males had a higher degree of slump sitting than females. These authors also observed that female adolescents reported BP made worse by sitting more frequently than male adolescents. Regarding age, it was found that 10-year-old children presented greater lumbar kyphosis while sitting than 11-year-old students. In addition, 11-year-old children had significantly less pelvic retroversion than 8, 9- and 10-year-old groups. 

These results are consistent with those found in previous studies. For instance, Cil et al. (2005) [61] studied the development of sagittal spinal curves in the standing position with children and observed that lordosis accelerates its growth from 10 to 12 years of age. Likewise, Kamaci et al. (2015) [5] investigated sagittal spinal development in a sitting position with children and found a significant increase in lumbar lordosis with age. Therefore, it can be considered normal that younger children had greater lumbar kyphosis and LSA while sitting than older students. For the same reason, it was also found that children with normal lumbar kyphosis were significantly taller and weightier than those who had slight or moderate lumbar hyperkyphosis. 

On the other hand, Kamaci et al. [5] observed that thoracic kyphosis remained relatively stable and tended to decrease at 13 to 17 years of age, while Cil et al. [61] found that thoracic kyphosis decreased from 10 to 12 years of age. In this sense, the results of the current investigation showed that 11-year-old children had less thoracic kyphosis than 10-year-old students.

Furthermore, it was found that the kyphotic lumbar angle while sitting was correlated in 63.8% with the results of LSA in a slump sitting position. It might be due to the fact that spine and sacropelvic have an interdepended anatomical relation [62], and thus, sacral slope influences the lumbar curve [63] so that the flexed position of the hip while sitting generates a backward tilt and subsequently increases lumbar kyphosis [5].

Secondly, the current investigation aimed to explore the association of sagittal spine and pelvic tilt with BP in 8 to 12-year-old children. In this sense, it was observed that those who did not suffer from BP in any part of the back had a higher lumbar kyphosis or a greater LSA than children who had some type of BP in the previous year or week. Nevertheless, it does not mean that a higher lumbar kyphosis or LSA while sitting can prevent BP. In fact, either those who had or who did not have BP presented a mean LSA with a slight retroversion or posterior pelvic tilt (101° to 110°), as well as both sufferers and non-sufferers had mean kyphotic lumbar angles which were categorized as slight lumbar hyperkyphosis (16° to 25°). In this sense, all prediction models had a poor accuracy to predict BP through LSA or lumbar angle.

Similar to this study, O’Sullivan et al. [60] found that there was no univariable association between BP made worse by sitting and the degree of the slump in sitting. The authors stated that BP made worse by sitting was weakly associated with more slump in sitting, whereas BP not made worse by sitting was weakly associated with less slump in sitting.

This controversial relationship could be justified through the multifactorial character of BP, the definition of back pain, the type of design used and the characteristics of the sample analyzed.

Smith et al. [64] indicate that not in all the situations the character of BP can be explained by studying the association of single and isolated measures of spinal posture. Thus, there are too many factors that have been associated with BP (demographic, socioeconomic, psychosocial, hereditary, anthropometric, behavioral, and postural factors, level of exercise, type of sport and intensity and frequency of BP) [65,66,67].

With respect to the sitting posture and BP, several studies that analyze the BP prevalence between students or athletes with and without BP show the importance to consider the degree and frequency of BP. In this sense, Noll et al. [66] compared different intensities and frequencies of BP in young athletes and found that behavioral and postural factors were associated with a high intensity and frequency of BP. Results of their multivariable analysis showed an association between high BP intensity and time spent using a computer (PR: 1.15, CI: 1.01–1.33), posture while writing (PR: 1.41, CI: 1.27–1.58), and posture while using a computer (PR: 1.39, CI: 1.26–1.54). Multivariable analysis also revealed an association of high BP frequency with studying in bed (PR: 1.19, CI: 1.01–1.40) and the method of carrying a backpack (PR: 1.19, CI: 1.01–1.40) and concluded that behavioral and postural factors are associated with a high intensity and frequency of BP. 

Hakala et al. [68] reported that frequent computer-related activities are a risk factor for neck, shoulder, and low back pain. Studies of college students demonstrated that periods in a sustained sitting posture and increased back flexion from sitting are significantly associated with BP [27]. Additionally, studies of workers revealed a positive association between the time spent sitting and BP [69]. Long-term sitting increases compression on the intervertebral discs, which leads to disc malnutrition, and it may compromise the integrity of the musculoskeletal system [70].

In addition, there has been found a correlation between BP and the number of hours the students dedicate to watch television [71,72,73,74], or with incorrect sitting posture due to inappropriate furnishing in the schools [75]. Similarly, students who remained seated for long periods throughout the day in an inappropriate posture (e.g., an unaligned head position, hyperlordotic or slumped trunk, and unaligned shoulders) [65,76], are predisposed to fatigue and higher levels of pain [63]. 

For that reason, and considering the relationship between back pain and sitting posture, it is necessary to carry out more studies that clarify the relationship and allow us to propose morphotypes and ranges of curvatures that predispose to pain.

On the other hand, it is necessary to take into account that the characteristics of the sample can influence the results (age range, sex, inclusion criteria, etc.). In this sense, in the current research, the students who had suffered an important physical injury which limited the correct performance of the test or who had been previously diagnosed to have scoliosis or who had previously received treatment for any frontal or sagittal plane-related pathology were excluded. Maybe, if all the students had been included, the results would have shown more correlation between the sagittal spinal curvatures and back pain, especially in those children with pathology of the sagittal plane of the spine. Some studies have found a relationship between different morphotypes of the spine in the sagittal plane and back pain or pathology [77], highlighting that in all of them, the correct or neutral posture has been related to the absence of pain.

In the current study, the percentages of children who indicated having suffered back pain were 10.7% for pain in the previous week and 22.3% in the previous year. However, the percentage of schoolchildren with good posture is small (only 44.50% of students presented a normal thoracic kyphosis, 12.54%, had a normal lumbar kyphosis while sitting, as well as only 14.78% of students presented a normal pelvic disposition). 

The percentages of back pain in the current study are in agreement with what is found in the literature for this age range [48,65,74,78]. Although, it should be considered that there are only a small number of studies that analyze the prevalence of back pain in schoolchildren aged 10 years or younger. LBP often begins in childhood, and in adolescents, the prevalence is similar to that of adults. One characteristic of LBP in childhood and adolescence is its high recurrence and the tendency to reappear with greater intensity. Although initially, intensity is usually low and it generally lasts for less than a week [48].

Finally, it must be taken into account that the type of design (cross-sectional study) may also influence the results. In this sense, Junge et al. [79] indicate that the prevalence studies in cross-sectional studies present the proportion of the population reporting BP at a certain time point, within a certain period or ever, and it is not possible to determine from these studies whether it is the same or different children or adolescents, who report BP at different ages and time points, seen in a long-term perspective [80]. As prevalence studies of BP only describe the population-averaged status of BP, and hence does not reflect the development or course of BP, they provide limited information about the condition with respect to portrayal of health care consequences and prevention strategies. 

For it, and taking the natural course of recurrent and fluctuating seen in longitudinal studies, the etiology and development of BP in children and adolescents and their relation with the posture also needs to be reflected in a long-term course. It also has to be mentioned that the BP questionnaire had to be filled out by parents according to the responses of their children to assure that all the form was correctly completed as well as to promote parent’s awareness of the back pain status of their children [81]. To note, the use of parental reports is very important with early school-age children (under 11 years old) because it helps to improve the quality of the information gathered and has been recommended in prior studies [82].

The sample size, using validated questionnaires for the Spanish population, the assessment protocol and data from elementary schoolchildren are the main strengths of this study. However, the cross-sectional design and the fact that the measurements were based on self-report and collected in a short-medium period are the main limitations.

## 5. Conclusions

The results of the present study demonstrate that incorrect posture while sitting is very frequent in young schoolchildren and it greatly affects sagittal spinal curvatures. Only 44.50% of students presented a normal thoracic kyphosis, as well as only 12.54% had normal lumbar kyphosis while sitting. The analysis of LSA indicated that only 14.78% of students presented a normal pelvic disposition, while 68.21% of children had a slight pelvic retroversion. A higher percentage of the girls presented normal values for all the parameters studied, compared to the boys. These results are very relevant, as students normally remain seated for many hours every day. It is important to note that the spine of schoolchildren is still in development, and a correct posture is necessary for its proper growth. 

However, no relationship has been found between back pain and sitting sagittal curvatures. In this sense, all prediction models had a poor accuracy to predict BP through LSA or lumbar angle.

For this reason, and taking into account the relationship found in other studies between back pain and sitting posture, it is necessary to carry out more studies that clarify the relationship and allow us to propose morphotypes and ranges of curvatures that predispose to pain.

It must be considered that the cross-sectional design of this study limits any causal conclusions, thus, prospective and intervention studies are needed to further study the relationship between BP and sagittal spinal alignment in a sitting posture.

## Figures and Tables

**Figure 1 ijerph-17-02578-f001:**
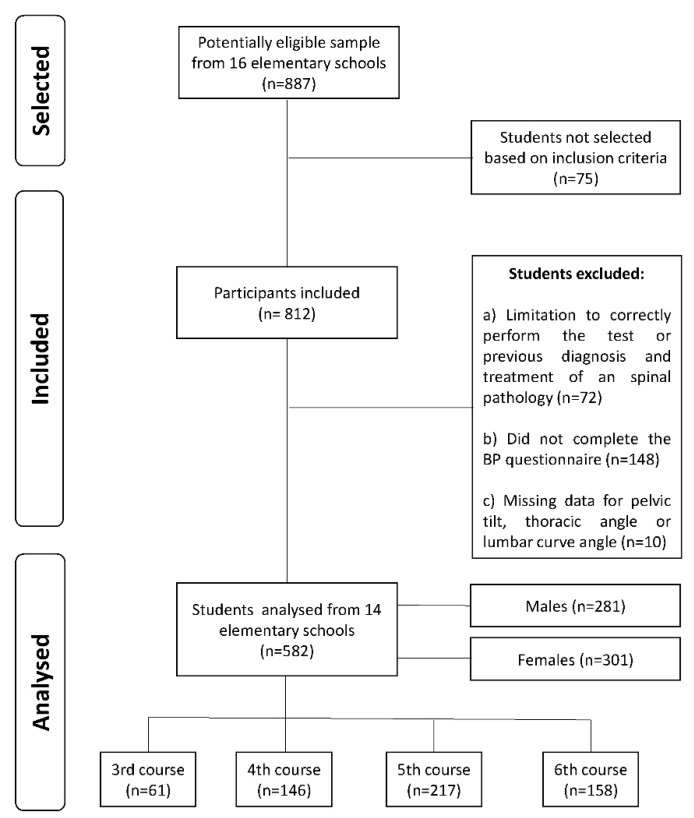
Flow diagram for the sample selection.

**Figure 2 ijerph-17-02578-f002:**
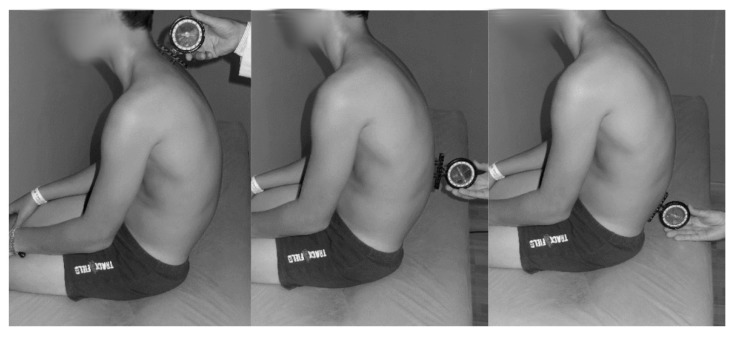
Assessment of thoracic and lumbar curves in a slump sitting position.

**Figure 3 ijerph-17-02578-f003:**
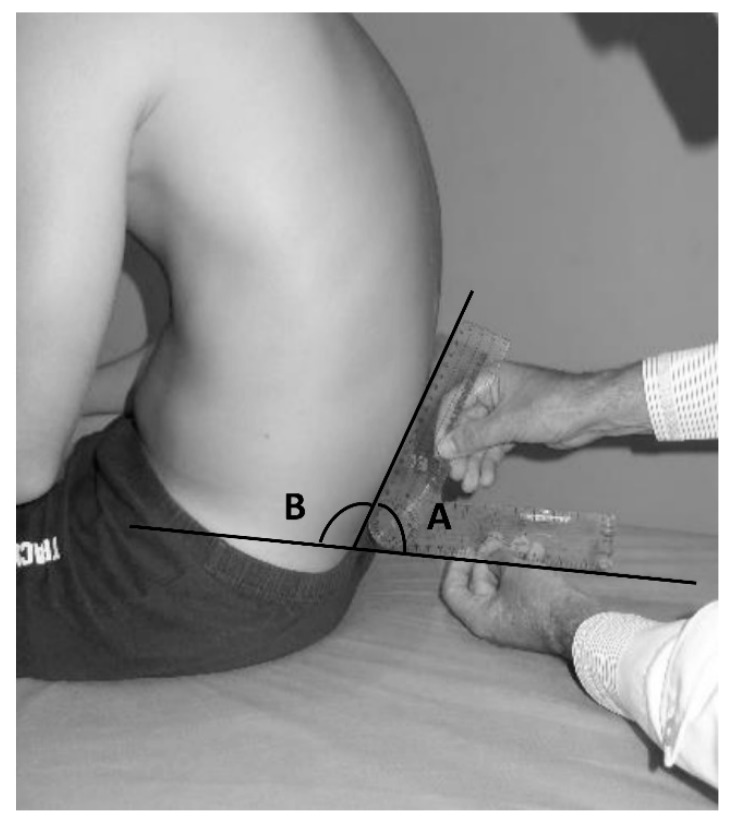
Measurement of the lumbosacral angle (LSA) in a slump sitting position. (**A**) recorded angle; (**B**) supplementary angle.

**Figure 4 ijerph-17-02578-f004:**
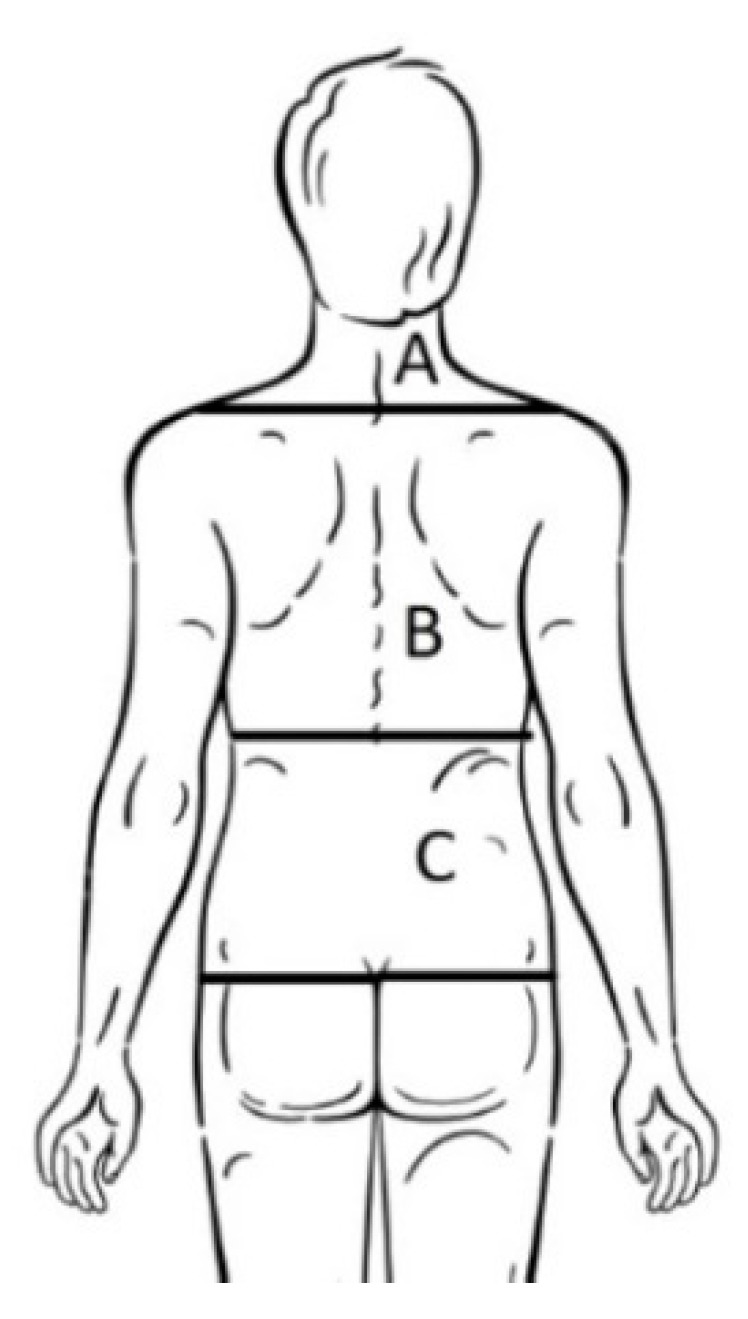
Drawing of the back to mark the back area where the pain was suffered.

**Figure 5 ijerph-17-02578-f005:**
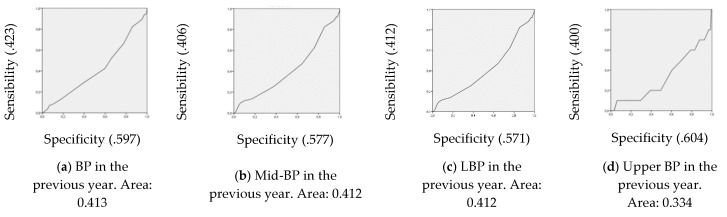
Sensibility, specificity and area under the receiver operating characteristic (ROC) curve for each analysis.

**Table 1 ijerph-17-02578-t001:** Demographics data of the final sample.

Variables		Age		Height		Weight	
		Mean	SD	Mean	SD	Mean	SD
Sex	Male (n = 281)	10.41	1.16	141.88	8.90	39.93	10.54
	Female (n = 301)	10.44	1.09	142.60	9.31	41.68	12.06
Age	8 years (n = 62)	8.44	0.29	129.70	5.21	30.82	6.01
	9 years (n = 132)	9.49	0.29	137.40	6.38	37.04	8.62
	10 years (n = 207)	10.49	0.28	142.80	7.08	40.19	9.46
	11 years (n = 142)	11.50	0.29	148.90	6.94	47.16	12.71
	12 years (n = 39)	12.47	0.34	151.57	6.34	50.00	11.26
	Total (n = 582)	10.42	1.12	142.25	9.11	40.84	11.37

SD: Standard deviation.

**Table 2 ijerph-17-02578-t002:** Angular values for thoracic curve, lumbar curve and pelvic tilt by sex and age.

Variable		Thoracic Curve ^1^	Lumbar Curve ^1^	Pelvic Tilt (LSA) ^1^
Sex	Male (n = 281)	43.86 ± 8.64 *	25.90 ± 7.42 †	107.99 ± 5.10 †
	Female (n = 301)	41.61 ± 9.83	22.79 ± 8.07	105.85 ± 5.79
Age	8 years (n = 62)	41.61 ± 7.83	25.81 ± 6.13	108.45 ± 4.63 †
	9 years (n = 132)	42.09 ± 9.51	24.68 ± 7.09	107.50 ± 5.02 †
	10 years (n = 207)	44.13 ± 9.30 *	25.05 ± 7.89 *	107.53 ± 5.45 †
	11 years (n = 142)	40.72 ± 9.71 *	22.62 ± 8.79 *	104.92 ± 6.10 †
	12 years (n = 39)	46.10 ± 7.93 *	22.62 ± 8.78	106.05 ± 5.55
	Total (n = 582)	42.70 ± 9.34	24.29 ± 7.91	106.88 ± 5.57

^1^ Thoracic, lumbar and pelvic angles are shown as mean ± standard deviation (SD); * *p* < 0.05; † *p* < 0.001.

**Table 3 ijerph-17-02578-t003:** Thoracic kyphosis, lumbar kyphosis and pelvic tilt according to normality references by sex and age. Comparison of height and weight among thoracic, lumbar and pelvic categories according to normality references.

V	Thoracic Kyphosis ^1^				Lumbar Kyphosis ^1^			Pelvic Tilt (Retroversion) ^1^		
	Hypo	N	S Hyper	M Hyper	N	S Hyper	M Hyper	N	S	Sig.
♂	1 (16.67)	112 (43.24)	164 * (52.73)	4 (66.67)	22 (30.14)	93 (41.33)	166 ^†^ (58.45)	28¥ (32.56)	197 (49.62)	56 (56.57)
♀	5 (83.33)	147 * (56.76)	147 (47.27)	2 (33.33)	51^†^ (69.86)	132 ^†^ (58.67)	118 (41.55)	58 * (67.44)	200 (50.38)	43 (43.43)
8y	0 (0.00)	32 (12.36)	30 (9.65)	0 (0.00)	2¥ (2.74)	25 (11.11)	35 (12.32)	5 (5.81)	47 (11.84)	10 (10.10)
9y	2 (33.33)	59 (22.78)	70 (22.51)	1 (16.67)	12 (16.44)	56 (24.89)	64 (22.54)	13 (15.12)	96 (24.18)	23 (23.23)
10y	2 (33.33)	74 (28.57)	127 (40.84)	4 (66.67)	21 (28.77)	77 (34.22)	109 (38.38)	24 (27.91)	141 (35.52)	42 (42.42)
11y	2 (33.33)	79 (30.50)	60 (19.29)	1 (16.67)	32 * (43.84)	47 (20.89)	63 (22.18)	35 * (40.70)	91 (22.92)	16¥ (16.16)
12y	0 (0.00)	15 (5.79)	24 (7.72)	0 (0.00)	6 (8.22)	20 (8.89)	13 (4.58)	9 (10.47)	22 (5.54)	8 (8.08)
T	6 (1.03)	259 (44.50)	311 (53.44)	6 (1.03)	73 (12.54)	225 (38.66)	284 (48.80)	86 (14.78)	397 (68.21)	99 (17.01)
H	146.92 ± 11.21	142.36 ± 9.12	141.97 ± 9.06	147.98 ±8.25	146.44 ± 8.30 *^,†^	142.49 * ± 8.71	140.99 ^†^ ± 9.31	146.67 ^†^ ± 9.03	141.72 ^†^ ± 8.79	140.57 ^†^ ± 9.38
W	46.37 ± 18.15	41.69 ± 12.05	40.01 ± 10.61	41.12 ±10.44	51.34^†^ ± 13.95	42.10 ^†^ ± 10.43	37.13 ^†^ ± 9.30	52.84 ^†^ ± 13.85	39.41 ^†,^* ± 9.43	36.14 ^†^ * ± 9.16

^1^ Values are represented as “Mean ± SD” or “n (%)”; * *p* < 0.05; † *p* < 0.001; ¥ significant and negative association; V = variables; ♂ = male sex; ♀ = female sex; y = years; N = normal; S = slight; M = moderate; Sig. = significant; T = total sample; H = height; W = weight.

**Table 4 ijerph-17-02578-t004:** Pelvic tilt by spinal curve according to normality references.

Spinal Curves		LSA or Pelvic Tilt		
		Normal ^1^	Slight Retroversion ^1^	Significant Retroversion ^1^
Thoracic kyphosis	Hypokyphosis	3 (3.49) ^†^	3 (0.76)	0 (0.00)
	Normal	50 (58.14) ^†^	186 (46.85)	23 (23.23) ^†^ ¥
	Slight hyperkyphosis	33 (38.37) ^†^ ¥	203 (51.13)	75 (75.76) ^†^
	Moderate hyperkyphosis	0 (0.00)	5 (1.26)	1 (1.01)
Lumbar kyphosis	Normal	40 (46.51) ^†^	32 (8.06) ^†^ ¥	1 (1.01) ^†^ ¥
	Slight hyperkyphosis	33 (38.37)	179 (45.09) ^†^	13 (13.13) ^†^ ¥
	Moderate hyperkyphosis	13 (15.12) ^†^ ¥	186 (46.85)	85 (85.86) ^†^

^1^ Values are presented as “n (%)”; ^†^
*p* < 0.001; ¥ negative association.

**Table 5 ijerph-17-02578-t005:** Angular value of thoracic curve, lumbar curve and pelvic tilt of those who had and who did not have back pain (BP) in the previous year and in the preceding week.

BP by Recurrence in the Previous Year, Body Area and Prevalence Period			Thoracic Curve ^1^	Lumbar Curve ^1^	Pelvic Tilt (LSA) ^1^
Previous year	BP	No (n = 452)	42.81 ± 9.23	24.64 ± 7.84 *	107.27 ± 5.38 *
		Yes (n = 130)	42.32 ± 9.73	23.08 ± 8.06	105.52 ± 6.00
		One-time event (n = 39)	43,54 ± 12,76	23,95 ± 7,29	106,21 ± 6,00
		Recurrent BP (n = 91)	41,80 ± 8,12	22,70 ± 8,38	105,23 ± 6,01
	Upper BP	No (n = 546)	42.65 ± 9.22	24.34 ± 7.94	106.91 ± 5.62
		Yes (n = 31)	43.16 ± 11.70	22.90 ± 7.28	105.94 ± 4.77
	Mid-BP	No (n = 508)	42.77 ± 9.60	24.45 ± 7.79	107.11 ± 5.39 *
		Yes (n = 69)	41.97 ± 7.35	22.87 ± 8.68	105.04 ± 6.57
	LBP	No (n = 525)	42.76 ± 9.28	24.44 ± 7.83	107.02 ± 5.45 *
		Yes (n = 51)	41.80 ± 10.29	22.47 ± 8.56	105.37 ± 6.58
Previous week	BP	No (n = 516)	42.62 ± 9.59	24.42 ± 7.89	106.97 ± 5.59
		Yes (n = 62)	43.48 ± 7.05	23.03 ± 8.14	106.29 ± 5.27
	Upper BP	No (n = 568)	42.70 ± 9.39	24.36 ± 7.86 *	106.93 ± 5.50
		Yes (n = 10)	43.60 ± 6.65	19.00 ± 10.08	105.20 ± 8.23
	Mid-BP	No (n = 533)	42.66 ± 9.54	24.38 ± 7.87	106.93 ± 5.55
		Yes (n = 45)	43.33 ± 6.77	22.93 ± 8.50	106.53 ± 5.70
	LBP	No (n = 561)	42.62 ± 9.38	24.35 ± 7.90	106.97 ± 5.54
		Yes (n = 17)	46.00 ± 7.87	21.41 ± 8.54	104.47 ± 5.50

^1^ Values are represented as mean ± SD; BP: back pain; LBP: low back pain; mid-BP: back pain in the middle back; * Significant differences by BP (Yes/No) (*p* < 0.05).

**Table 6 ijerph-17-02578-t006:** Relative frequencies and logistic regression results.

Variable	Categories	LSA		Lumbar Curve		OR *	SE	95% CI	*p*-Value
		<107°	≥107°	<23°	≥23°				
BP (year)	No (n = 516)	39.2%	60.8%	-	-	1.059 (small)	0.018	1.021 to 1.097	0.002
	Yes (n = 62)	28.5%	71.5%	-	-				
MB pain (year)	No (n = 508)	62.2%	37.8%	-	-	1.069 (small)	0.023	1.021 to 1.118	0.004
	Yes (n = 69)	71.0%	29.0%	-	-				
LBP (year)	No (n = 525)	62.1%	37.9%	-	-	1.054 (small)	0.026	1.001 to 1.109	0.045
	Yes (n = 51)	74.5%	25.5%	-	-				
UB pain (week)	No (n = 568)	-	-	49.3%	50.7%	1.082 (small)	0.026	1.005 to 1.116	0.036
	Yes (n = 10)	-	-	20.0%	80.0%				

BP: back pain; MB: middle back; LBP: low back pain; UB: upper back; OR: odds ratio (relative risk); SE: standard error; CI: confidence interval. * OR < 1: poor predictor of LBP; OR from 1 to 1.25: small predictor; OR from 1.25 to 2: medium predictor; OR ≥ 2: large predictor [50].

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
