# Peer review of "Sitting Posture, Sagittal Spinal Curvatures and Back Pain in 8 to 12-Year-Old Children from the Region of Murcia (Spain): ISQUIOS Programme"

_ijerph, 2020, doi:10.3390/ijerph17072578_

Round 1
Reviewer 1 Report
Dear Authors:
This manuscript addresses an important public health issue, i.e. the analysis of the sagittal spinal and the relation with back pain in school students. The manuscript is of interested and the methodology is appropriate; however, there are a few minor issues that should be addressed by the authors to improve the manuscript. See the comments below.
Abstract
Line 29. Indicate acronym BP.
Introduction
Maybe, there are a lot of references with 20 years ago.
Method
Desig. You write “In addition, the students filled out a questionnaire about the BP suffered in the previous week and in the preceding year”. But, in the point 2.3.3. you describe the “back pain assessment”for students and parent.
And in the results, we don't know to which collective does it refer.
Results
It is good.
Conclussion
Maybe, the conclussion are very long and it repeated de results. You could be more specific.
References
The references have a lot problem.
You only have 8 references with less than 5 years. Approximately 9-10%.
With 5 to 10 years, the references are appoximately 22%
But the big problem is that you have more than 65% of the references, with more than 10 years. With a large volume of more than 20 years.
The article is well written, structured and interesting. But the bibliography is very old. They should update it, and indicate the references of the 90s, only those that are exclusively necessary.
Thanks to the authors for their effort.
Author Response
Response to Reviewer 1 Comments
Dear Fiona Zhao and reviewer 1,
Thank you for the comments and suggestions proposed for the article “Sitting posture, Sagittal Spinal Curvatures and Back Pain in 8 to 12-Year-Old Children from the Region of Murcia (Spain): ISQUIOS Programme”.
We are pleased to provide a cover letter to explain *point-by-point* the
details of the revisions in the manuscript and our responses to the
reviewers' comments. Any revisions are “clearly highlighted” with the “Track Changes” function in Microsoft Word.
Explanation point by point:
Point 1: Abstract. Line 29. Indicate acronym BP.
Response 1: Thank you for your suggestion. The change has been made.
Point 2: Introduction. Maybe, there are a lot of references with 20 years ago.
Response 2: Thank you for your suggestion. The references have been updated.
Point 3: Method. Design. You write “In addition, the students filled out a questionnaire about the BP suffered in the previous week and in the preceding year”. But, in the point 2.3.3. you describe the “back pain assessment”for students and parent. And in the results, we don't know to which collective does it refer.
Response 3: As you indicate, it is not clear who we are referring to. So, to avoid confusion, it has been specified in the point 2.3.3 that the questionnaire has been used to determine the prevalence of BP in schoolchildren, but since they were minors, parents filled it out for them.
Point 4: Conclussion. Maybe, the conclussion are very long and it repeated the results. You could be more specific.
Response 4: Thank you for the suggestion. The conclusion has been summarized and modified in order to focus more on the take-home message and less on the repetition of results.
Point 5: References. The references have a lot problem. You only have 8 references with less than 5 years. Approximately 9-10%. With 5 to 10 years, the references are appoximately 22%. But the big problem is that you have more than 65% of the references, with more than 10 years. With a large volume of more than 20 years.
Response 5: Thank you for your suggestion. All the references have been reviewed and updated according to your recommendations.
Point 6: The article is well written, structured and interesting. But the bibliography is very old. They should update it, and indicate the references of the 90s, only those that are exclusively necessary.
Response 6: Thank you for your suggestion. All the references have been reviewed and updated according to your recommendations.
We hope that all the changes can address your considerations.
Kind regards,
The authors.

Reviewer 2 Report
The goal of the present study was to describe sagittal spinal alignment and pelvic tilt in a slump sitting position, and to explore the association of sagittal spine and pelvic tilt with BP among 8-12 year-old children.
The authors took on a huge role to physically examine almost 600 students and gather questionnaires on back pain. However, there are a few major methodological flaws I strongly suggest to correct:
The study design is a cross sectional study; since exposures and outcomes are measured at the same time, only prevalence of outcomes and exposures can be presented, and hypotheses can be generated. Therefore, any kind of association measures such as odds ratios should not be calculated. Of note, the odds ratios presented in table 5 are outside the 95% confidence interval, which cannot be correct.
The questionnaire on back pain asked for the one-week and one-year prevalence of back pain. The one-year prevalence may be biased by recall, which should be discussed. It is not clear who completed the questionnaire as it is stated both - children (line 90) and parents (line 158). I am wondering whether the parents marked the affected areas on the back. Outcomes may differ if measured by proxy or the children themselves. This may need to be discussed as back pain is one main outcome of the study. I am wondering whether pain was evaluated as a one time event or if recurrent pain was taking into account. A one time event may have a number of causes, which may need to be discussed.
I am wondering - is the reason for the back pain different when measured in this age group as opposed to the back pain in adolescents or adult. Would the hypotheses differ in both age groups: An appropriate "neutral" posture may cause pain due to muscular fatigue in young children versus a long-standing slump sitting position may lead to back pain in adolescents/ adults - possibly due to degenerative disease? These two concepts may need to be clarified for this study.
The authors should describe the "convenience sample" a little more in detail. It may need to be discussed how the sample potentially differs from the general population of students of this age group.
Outliers of sagittal spinal alignment and pelvic tilt were excluded - how would the results change if they weren't? Were the outliers those who had pain?
The method part contains some results such as number of included students, number of excluded students with reasons, and demographics. The authors may consider to present a flow diagram or just description of excluded students and a table on demographics as part of the results.
The discussion part would benefit from a clear outline about what is new, and what are the limitations and strengths of the study. The conclusion may focus more on the take-home message and less on the repetition of results.
Some general recommendations:
The paper is quite long and repetitive (particularly comparing introduction and discussion) and would benefit from a more concise reporting style.
I am not sure whether all statistics really apply to the study, and strongly suggest to receive statistical advice that should include the presentation of the data.
I am not a native English speaker; however, I am wondering if the manuscript should be reviewed by an English editor - it may help with shortening of the manuscript as well.
Author Response
Response to Reviewer 2 Comments
Dear Fiona Zhao and reviewer 2,
Thank you for the comments and suggestions proposed for the article “Sitting posture, Sagittal Spinal Curvatures and Back Pain in 8 to 12-Year-Old Children from the Region of Murcia (Spain): ISQUIOS Programme”.
We are pleased to provide a cover letter to explain *point-by-point* the
details of the revisions in the manuscript and our responses to the
reviewers' comments. Any revisions are “clearly highlighted” with the “Track Changes” function in Microsoft Word.
Explanation point by point:
Comments and Suggestions for Authors
Point 1: The study design is a cross sectional study; since exposures and outcomes are measured at the same time, only prevalence of outcomes and exposures can be presented, and hypotheses can be generated. Therefore, any kind of association measures such as odds ratios should not be calculated. Of note, the odds ratios presented in table 5 are outside the 95% confidence interval, which cannot be correct.
Response 1: First of all, thanks for your helpful consideration. In this case, odds ratios were calculated as a measure of the power of the association that was cross-sectionally found. Thus, it shows the power of the cross-sectional relationship between LSA or lumbar curve’s values and back pain. Therefore, the explanation which is within the results section above table 5 has been modified to clarify that no causal relationship has been established. In addition, the 95% confidence intervals for the odds ratios have been corrected in the text and within table 5. We hope that these changes can address your considerations.
In addition, in the discussion section we have commented on the influence that the type of design may have had on the results.
Point 2: The questionnaire on back pain asked for the one-week and one-year prevalence of back pain. The one-year prevalence may be biased by recall, which should be discussed. It is not clear who completed the questionnaire as it is stated both - children (line 90) and parents (line 158). I am wondering whether the parents marked the affected areas on the back. Outcomes may differ if measured by proxy or the children themselves. This may need to be discussed as back pain is one main outcome of the study. I am wondering whether pain was evaluated as a one time event or if recurrent pain was taking into account. A one time event may have a number of causes, which may need to be discussed.
Response 2: In relation to the one-year prevalence, it is true that the answer may be biased by the recall, so asking also for pain in other times frames can help for the comparability and so, remember better the real back pain.
Regarding the questionnaire, as you indicate, it is not clear who completed it. So, to avoid confusion, it has been specified in the point 2.3.3 that the questionnaire has been used to determine the prevalence of BP in schoolchildren, but since they were minors, parents filled it out for them.
Finally, a definition of back pain is presented in the questionnaire, to avoid indicating pain due to feverish illness or menstruation. This definition was also explained to the parents when the questionnaire was delivered. In addition, questions about the frequency and duration of back pain are included. However, these data were not used in the present study.
The definition of back pain in our study was evaluated as a one time event. In the discussion section we discus about the causes of back pain as a one time event.
In relation to the recurrent LBP, we took into account the Staton et al. (2010) research about how to define the condition of “recurrent low back pain”.
The firts published recommended definition for recurrent LBP was proposed by von Korff in 1994: ‘back pain present on less than half the days in a 12-month period, occurring in multiple episodes over the year’ [Korff, 1994].
While Staton et al. (2010) suggest that the following features should be included as part of a definition of recurrent LBP:
- A) To provide a definition of an episode of LBP that includes a definition for the start and end of the episode and advocate the de Vet definition (period of LBP lasting more than 24 h preceded and separated by a period of at least 1 month without LBP) [Vet et al., 2002] as suitable for this purpose.
- B) A definition to classify someone as having recurrent LBP needs to consider the number of previous episodes of LBP and the time span they occurred over (e.g., at least 2 episodes in the past 12 months). These recommendations reflect the opinions of the authors based upon review of the current literature.
Korff M. Studying the natural history of back pain. Spine. 1994;19:2041S–2046S. doi: 10.1097/00007632-199409151-00005.
Vet HCW, Heymans MW, Dunn KMP DP, Beek AJ, Macfarlane GJ, Bouter LM, Croft PR. Episodes of low back pain: a proposal for uniform definitions to be used in research. Spine. 2002;27:2409–2416. doi: 10.1097/00007632-200211010-00016.
Point 3: I am wondering - is the reason for the back pain different when measured in this age group as opposed to the back pain in adolescents or adult. Would the hypotheses differ in both age groups: An appropriate "neutral" posture may cause pain due to muscular fatigue in young children versus a long-standing slump sitting position may lead to back pain in adolescents/ adults - possibly due to degenerative disease? These two concepts may need to be clarified for this study.
Response 3: Thank you for your suggestion. This fact has been explained in the “Discussion” section.
Point 4: The authors should describe the "convenience sample" a little more in detail. It may need to be discussed how the sample potentially differs from the general population of students of this age group.
Response 4: The convenience sample refers to the fact that their elementary schools were selected to participate in the ISQUIOS programme and offered voluntarily to participate in this study. The sample does not differ from the general population of this age group because the assessments were made before the intervention program began. This fact has been explained in the “method” section.
Point 5: Outliers of sagittal spinal alignment and pelvic tilt were excluded - how would the results change if they weren't? Were the outliers those who had pain?
Response 5: Outliers were removed from the sample due to a mistake with respect to the measure and data collection. Therefore, the influence of these cases on the back-pain results cannot be studied.
Point 6: The method part contains some results such as number of included students, number of excluded students with reasons, and demographics. The authors may consider to present a flow diagram or just description of excluded students and a table on demographics as part of the results.
Response 6: A flow diagram for the sample selection and a table on participants demographics as part of the results have been included.
Point 7: The discussion part would benefit from a clear outline about what is new, and what are the limitations and strengths of the study. The conclusion may focus more on the take-home message and less on the repetition of results.
Response 7: Thank you for your suggestion. The discussion part has been modified and we have added the limitations and strengths of the study.
Point 8: Some general recommendations:
The paper is quite long and repetitive (particularly comparing introduction and discussion) and would benefit from a more concise reporting style.
I am not sure whether all statistics really apply to the study, and strongly suggest to receive statistical advice that should include the presentation of the data.
I am not a native English speaker; however, I am wondering if the manuscript should be reviewed by an English editor - it may help with shortening of the manuscript as well.
Response 8: Thank you for your suggestions. We have received statistical advice and the writing has been modified to make the manuscript clearer and to improve the English.
We hope that all the changes can address your considerations.
Kind regards,
The authors.

Round 2
Reviewer 2 Report
Thanks to the authors for all their work on improving the manuscript and address my concerns.
I still have a few concerns: I am wondering about the importance of one-time back pain. Is there any clinical implication and/ or significance if a child has one time back pain. If you have the data on frequency of back pain could you use them? It may give you a completely different and possibly more meaningful outcome if you can analyse the data for recurrent back pain.
Other concerns may need to be addressed:
In the era of patient - including children - engagement in studies I strongly suggest not to use the explanation "children were minor" why parents completed the questionnaires. I would just state it and explain the potential biases associated with it in the discussion part.
If there were mistakes coming up in the measurements leading to exclusion of patients: As it was the same person performing the measurements, were they random? how did you maintain quality? Was there any double data measurement or entry for data validation?
The reason why "outliers" were excluded have to be explained in the text. It has to be clarified whether it is experimental error or random measurements. The diagram does not refer to outliers and it remains therefore unclear.
The manuscript is still long and will benefit from shortening.
Author Response
Response to Reviewer 2 Comments
Dear reviewer 2,
Thank you for the comments and suggestions proposed for the article “Sitting posture, Sagittal Spinal Curvatures and Back Pain in 8 to 12-Year-Old Children from the Region of Murcia (Spain): ISQUIOS Programme”.
We are pleased to provide a cover letter to explain *point-by-point* the details of the revisions in the manuscript and our responses to the comments. Any revisions are “clearly highlighted” with the “Track Changes” function in Microsoft Word.
Explanation point by point:
Comments and Suggestions for Authors
Thanks to the authors for all their work on improving the manuscript and address my concerns.
Point 1: I still have a few concerns: I am wondering about the importance of one-time back pain. Is there any clinical implication and/ or significance if a child has one time back pain. If you have the data on frequency of back pain could you use them? It may give you a completely different and possibly more meaningful outcome if you can analyse the data for recurrent back pain.
Response 1: Thank you again for offering us the possibility to improve this paper. Following your recommendation, the analysis of spinal curvatures by back pain recurrence in the previous year has been included within table 5. In addition, a T-student test for independent samples was used to compare spinal and pelvic means between those who suffered the back pain once or more than once, and no difference was found among groups (p>0.05). However, we would like to outline that there are clinical implications if a child experience one-time back pain. For instance, some studies have found that BP prevalence tends to increase with age and those children who suffered from BP at an early age were more likely to experience BP in their adulthood (Aparicio-Sarmiento et al., 2019; Burton et al., 1996; Calvo-Muñoz et al., 2013; Jeffries et al., 2007; Kjaer et al., 2011; Martínez-Crespo et al., 2009; Miñana-Signes & Monfort-Pañego, 2015; Silva et al., 2014; Yao et al., 2011). In addition, this high prevalence in children causes an increase in the number of doctor visits and the general use of the public health services (Kjaer et al., 2011; Myrtveit et al., 2014). This means that even if it is a one-time event, BP needs to be considered because it can affect children’s physical health, emotional functioning and psychosocial health and result in a decrease of their quality of life (Macedo et al., 2015). This scenario is more worrisome if we take into account that the problem of BP is increasing in childhood and adolescence as the most recent studies tend to show higher prevalence rates (Calvo-Muñoz et al., 2013).
To note, most studies about BP have been carried out in older populations. For that reason, prior to address the problem of BP in young children, it is important to know what the associated factors to BP in this young population are. Thus, the current study investigates some factors and their relationship with BP to obtain a better understanding of the BP issue in a large sample of young schoolchildren.
Aparicio-Sarmiento, A., Rodríguez-Ferrán, O., Martínez-Romero, M. T., Cejudo, A., Santonja, F., & Sainz de Baranda, P. (2019). Back Pain and Knowledge of Back Care Related to Physical Activity in 12 to 17 Year Old Adolescents from the Region of Murcia (Spain): ISQUIOS Programme. Sustainability, 11(19), 5249. https://doi.org/10.3390/su11195249
Burton, A. K., Clarke, R. D., Mc Clune, T. D., & Tillotson, K. M. (1996). The Natural History of Low Back Pain in Adolescents. Spine, 21(20), 2323–2328.
Calvo-Muñoz, I., Gómez-Conesa, A., & Sánchez-Meca, J. (2013). Prevalence of low back pain in children and adolescents: A meta-analysis. Biomed Central Pediatrics, 13(14), 1–12.
Jeffries, L. J., Milanese, S. F., & Grimmer-Somers, K. A. (2007). Epidemiology of adolescent spinal pain: A systematic overview of the research literature. Spine, 32(23), 2630–2637.
Kjaer, P., Wedderkopp, N., Korsholm, L., & Leboeuf-Yde, C. (2011). Prevalence and tracking of back pain from childhood to adolescence. BMC Musculoskeletal Disorders, 12(98), 1–11. https://doi.org/10.1186/1471-2474-12-98
Macedo, R. B., Coelho-e-Silva, M. J., Sousa, N. F., Valente-dos-Santos, J., Machado-Rodrigues, A. M., Cumming, S. P., Lima, A. V., Gonçalves, R. S., & Martins, R. A. (2015). Quality of life, school backpack weight, and nonspecific low back pain in children and adolescents. Jornal de Pediatria, 91(3), 263–269. https://doi.org/10.1016/j.jped.2014.08.011
Martínez-Crespo, G., Rodríguez-Piñero, M., López-Salguero, A. I., Zarco-Periñan, M. J., Ibáñez-Campos, T., & Echevarría-Ruiz de Vargas, C. (2009). Dolor de espalda en adolescentes: Prevalencia y factores asociados. Rehabilitación (Madrid), 43(2), 72–80.
Miñana-Signes, V., & Monfort-Pañego, M. (2015). Back Health in Adolescents between 12-18 Years of the Valencian Community, Spain: Prevalence and Consequences. Journal of Spine, 4(4), 1–5.
Myrtveit, S. M., Sivertsen, B., Skogen, J. C., Frostholm, L., Stormark, K. M., & Hysing, M. (2014). Adolescent Neck and Shoulder Pain—The Association With Depression, Physical Activity, Screen-Based Activities, and Use of Health Care Services. Journal of Adolescent Health, 55(3), 366–372. https://doi.org/10.1016/j.jadohealth.2014.02.016
Silva, M. R. O. G. C. M., Badaró, A. F. V., & Dall’Agnol, M. M. (2014). Low back pain in adolescent and associated factors: A cross sectional study with schoolchildren. Brazilian Journal of Physical Therapy, 18(5), 402–409.
Yao, W., Mai, X., Luo, C., Ai, F., & Chen, Q. (2011). A cross-sectional survey of nonspecific low back pain among 2083 schoolchildren in China. Spine, 36(22), 1885–1890. https://doi.org/10.1097/BRS.0b013e3181faadea
Other concerns may need to be addressed:
Point 2: In the era of patient - including children - engagement in studies I strongly suggest not to use the explanation "children were minor" why parents completed the questionnaires. I would just state it and explain the potential biases associated with it in the discussion part.
Response 2: Thanks for the suggestion. The statement has been removed and it has been explained at the end of the discussion section as part of the study strengths and limitations.
Point 3: If there were mistakes coming up in the measurements leading to exclusion of patients: As it was the same person performing the measurements, were they random? how did you maintain quality? Was there any double data measurement or entry for data validation?
Response 3: Three trials for each measure were administered/recommended. When two of those measures were equal, we chose that value. When the three measures were different we took the average value of the two similar measurements for data analysis. Furthermore, it is very important to note that we made a test trial before the first measurement with the objective that the student was well informed and was sure about how to do the test. The measurements were made in a randomized order (Thoracic and lumbar sagittal curves and pelvic tilt). The study was rigorously controlled by keeping the expert and the students blinded to the objective of the study.
In addition, to assure quality and to establish the reliability of the examiner, a double-blind study with 10 subjects was performed before the assessment, and the results showed an Intraclass Correlation Coefficient (ICC) of 0.95 for the thoracic kyphosis and 0.95 for the lumbar lordosis.
Data validation is a decisional procedure ending with an acceptance or refusal of data as acceptable.
Point 4: The reason why "outliers" were excluded have to be explained in the text. It has to be clarified whether it is experimental error or random measurements. The diagram does not refer to outliers and it remains therefore unclear.
Response 4: With the term “outliers” we referred to those cases who presented missing data for some of the variables (pelvic tilt, thoracic angle or lumbar curve angle). It happened because some of those measures were not taken the day of the assessment for some reason, or because there was a mistake in data digitalization to databases. To clarify this point, it has been explained in the text and the diagram has been modified replacing “mistakes in the measure or data collection” for “missing data for pelvic tilt, thoracic angle or lumbar curve angle”. In addition, the term “outliers” has been corrected and change to “missing data”.
Point 5: The manuscript is still long and will benefit from shortening.
Response 5: Thank you for the suggestion. Summarize the manuscript is not easy and sometimes it can affect negatively to the understanding of the whole content. Maybe the discussion part could be shorter. However, in the last revision of the manuscript we adressed that consideration and we deleted some paragraphs and summarized the conclusion part to make it clearer and easy to read.
We hope that all the changes can address your considerations.
Kind regards,
The authors.
